# Molecular Mechanism of Anti-Inflammatory Activities of a Novel Sulfated Galactofucan from *Saccharina japonica*

**DOI:** 10.3390/md19080430

**Published:** 2021-07-29

**Authors:** Xiaodan Chen, Liying Ni, Xiaoting Fu, Lei Wang, Delin Duan, Luqiang Huang, Jiachao Xu, Xin Gao

**Affiliations:** 1College of Food Science & Engineering, Ocean University of China, 5th Yushan Road, Qingdao 266003, China; chenxiaodan@stu.ouc.edu.cn (X.C.); niliying@stu.ouc.edu.cn (L.N.); leiwang2021@ouc.edu.cn (L.W.); xujia@ouc.edu.cn (J.X.); xingao@ouc.edu.cn (X.G.); 2State Key Lab of Seaweed Bioactive Substances, Qingdao Bright Moon Seaweed Group Co., Ltd., 1th Daxueyuan Road, Qingdao 266400, China; dlduan@qdio.ac.cn; 3CAS and Shandong Province Key Lab of Experimental Marine Biology, Center for Ocean Mega-Science, Institute of Oceanology, Chinese Academy of Sciences, 7th Nanhai Road, Qingdao 266071, China; 4Key Laboratory of Special Marine Bio-Resources Sustainable Utilization of Fujian Province, College of Life Science, Fujian Normal University, Fuzhou 350108, China; biohlq@fjnu.edu.cn

**Keywords:** *Saccharina japonica*, galactofucan, structure, anti-inflammation, RAW 264.7, zebrafish

## Abstract

Seaweed of *Saccharina japonica* is the most abundantly cultured brown seaweed in the world, and has been consumed in the food industry due to its nutrition and the unique properties of its polysaccharides. In this study, fucoidan (LJNF3), purified from *S. japonica*, was found to be a novel sulfated galactofucan, with the monosaccharide of only fucose and galactose in a ratio of 79.22:20.78, and with an 11.36% content of sulfate groups. NMR spectroscopy showed that LJNF3 consists of (1→3)-α-l-fucopyranosyl-4-SO_3_ residues and (1→6)-β-d-galactopyranose units. The molecular mechanism of the anti-inflammatory effect in RAW264.7 demonstrated that LJNF3 reduced the production of nitric oxide (NO), and down-regulated the expression of MAPK (including p38, ENK and JNK) and NF-κB (including p65 and IKKα/IKKβ) signaling pathways. In a zebrafish experiment assay, LJNF3 showed a significantly protective effect, by reducing the cell death rate, inhibiting NO to 59.43%, and decreasing about 40% of reactive oxygen species. This study indicated that LJNF3, which only consisted of fucose and galactose, had the potential to be developed in the biomedical, food and cosmetic industries.

## 1. Introduction

Seaweed of *Saccharina japonica* is the most abundantly cultured brown seaweed in the world, and has been consumed in the food industry due to its nutrition and the unique properties of its polysaccharides. Polysaccharides that are found in *S. japonica* mainly refer to alginate, fucoidan, and laminarian, which have attracted widespread attention in biochemical and medical areas, offering different molecular structure characteristics and biological activities.

Fucoidan is a water-soluble polysaccharide with a sulfate group, and its main structural unit is α-1,3-linked or α-1,4-linked L-fucose residue. The monosaccharide composition of fucoidan is complex, mainly including fucose, galactose, mannose, fructose, xylose, and so on. Galactofucan is a novel fucoidan, containing only fucose and galactose in the monosaccharide composition, which have been found in brown seaweeds. Though there were numerous researches related to fucoidan, only a few galactofucans were reported to be purified from brown seaweeds, such as *Saccharina* [1,2,3,4,5,6,7,8], *Sargassum* [9,10,11,12,13,14,15,16,17,18], *Alaria* [19,20], *Costaria* [21], *Hizikia* [22], *Hormophysa* [23], *Kjellmaniella* [24], *Lobophora* [25], *Undaria* [26,27], *Scytosiphon* [28], and *Spatoglossum* [29,30], with different biological activities of anti-oxidant activity [25], anti-inflammatory [25], antiviral activity [2,11,28,31], antitumor activity [9,10,13,19], neuroprotective activity [14], and anti-angiogenic activity [12].

The present researches mainly studied the biological activities of galactofucan from *Saccharina* spp. and *Sargassum* spp. The researches of galactofucan from *Saccharina* spp. investigated the antiviral and antitumor activities, and explored the relationship between the structure and biological activities [2,4]. For example, it is reported that the sulfated galactofucan showed strong binding ability to SARS-CoV-2 spike glycoproteins, which may prevent and/or treat SARS-CoV-2 [2]. Besides, the galactofucan (MW123kDa) with 25.1% sulfate groups had no cytotoxicity in vitro and suppressed the colony formation of DLD-1 cells [4]. In addition, Jin et al. (2020) studied the structure–activity relationship of 68 types of marine oligosaccharides and polysaccharides, and found that sulfated galactofucans represent a good regulator of fibroblast growth factor 1 in cell development [3].

An excessive inflammatory response would lead to the release of a large number of pro-inflammatory cytokines, which have a serious impact on host cells and tissues, and then cause a variety of chronic inflammation [32]. Fucoidan was reported to show effective anti-inflammatory activity. There is a review about the anti-inflammatory effects of fucoidan, which summarized the molecular mechanism of fucoidan in vitro and in vivo [32]. However, there are no studies researching the anti-inflammatory activity of galactofucan derived from *S. japonica*. Therefore, it is necessary to explore the anti-inflammatory mechanism of galactofucan obtained from *S. japonica*, and evaluate the structural characteristics and biological activities of galactofucan.

In this study, we investigated the physicochemical properties of a purified fucoidan named LJNF3 (with a molecular weight of 261.7 KDa) from *S. japonica*, which was composed of fucose and galactose. The anti-inflammatory activity and mechanism of LJNF3 as galactofucan were studied by RAW 264.7 cells and zebrafish models in vitro and in vivo, and the relationship between the structure and anti-inflammatory activity was explored, so as to provide a theoretical basis for the role of LJNF3 as an anti-inflammatory component in pharmaceutical, functional food, and cosmetics.

## 2. Results and Discussion

### 2.1. Physicochemical Properties of Polysaccharide Fractions Isolated from S. japonica

The crude polysaccharide LJN from *S. japonica* was extracted by hot water, with a yield of 5.96 ± 1.35%, based on dried weight. After purification on a DEAE-Sepharose Fast Flow column (Figure 1), three purified fractions of polysaccharides were collected and named as LJNF1, LJNF2, and LJNF3, with yields of 9.59%, 55.88%, and 34.53%, respectively. As shown in Table 1, the chemical content, monosaccharide composition, and molecular weight of three fractions were determined. From the chemical content result, LJNF3 was composed of carbohydrates (69.12%) and sulfates (11.36%), without protein and phenol. The Mw of LJNF3 was 261.7 kDa, which was lower than those of the other two fractions.

Monosaccharide composition analysis indicated that fraction LJNF3 only constituted fucose (79.22%) and galactose (20.78%). The monosaccharide composition and biological activities of the reported galactofucans were summarized in Table 2. Galactofucans existed in many different genera of brown algae, especially in the genera of *Saccharina* and *Sargassum*. The galactofucans that are composed of only galactose and fucose are rare (underlined in Table 2). Though the reported fucoidans in Table 2 have all been named galactofucans in their literatures, most of them were composed of galactose and fucose, plus small amounts of other monosaccharides. In addition, Table 2 showed that the galactofucans obtained from different algae sources have certain differences in their structures, referring to the content of sulfate groups and molecular weight. However, a single universal relationship between the structure and activity cannot be established. A review has also discussed the relationship between the anti-inflammatory activity and molecular weight of fucoidan, indicating that low-molecular-weight fucoidan obtained by different depolymerization techniques showed some better pharmacological effects compared to high-molecular-weight ones. However, some very low-molecular-weight fractions (below 10 or 30 kDa) were reported as nonactive [32]. In Table 2, it can be found that galactofucans offer the activities of anti-tumor, antiviral and growth-promoting activity of fibrocytes. However, there was no study of the anti-inflammatory activity. As we know, inflammation is the cause of many chronic diseases, such as neurological diseases and chronic autoimmune diseases. Therefore, in this paper, the structure and anti-inflammatory mechanism of LJNF3 (galactofucan) was investigated.

### 2.2. Structural Feature of LJNF3

The IR spectra of LJNF3 was shown in Figure 2. A broad intense peak at 3480 cm^−1^ and signal at 1077 cm^−1^ represented the stretching vibration of O–H linkage, which indicated strong inter- and intra-molecular interactions of the polysaccharide chains. A narrow weak peak around 2942 cm^−1^ resulted from C–H stretching vibrations. The absorption band at 1260 cm^−1^ was related to S=O stretch. The signal at 964 cm^−1^, derived from the asymmetrical stretching vibration of C–O–S bonds, appeared in the spectra [37]. The range around 810–850 cm^−1^ represented the C–O–S stretching vibration, which showed the position of the sulfate groups. The absorption at 852 cm^−1^ was suggested to be due to sulfate groups at the axial C4 position [38].

Table 3 showed the structure of LJNF3 by 1D and 2D NMR spectroscopy. High-field (1.0–1.5 ppm) regions and α-anomeric (5.0‒5.6 ppm) were contained in 1H NMR (Figure 3a). The signals region had been assigned to the C6 methyl proton group of L-fucopyranose at 1.44 ppm and 1.02 ppm, and (1–3)-linked fucosyl residues at 1.2 ppm [2,39]. Between 3.6 and 4.8 ppm, the spectrum contained characteristic resonances of ring protons (H2–H5) and it was attributed to the H-4 of 4–O-sulfated residues at the signal region 4.78 ppm [40]. Besides, there was a correlation peak in the anomeric region from the two-dimensional 1H/1H COSY NMR spectrum (Figure 3c), which revealed the presence of one dominant constituent of L-fucopyranosyl residues A. The H6/H5 correlations at 1.32/4.12 ppm and 1.26/4.51 ppm were assigned to 1,3-linked α-l-fucopyranose units. The position of H1/H2 cross-peaks were 15.05/3.93 ppm related to the 4-sulfated fucosyl residue [41]. Besides, 3.84/65.75 and 3.66/60.96 ppm of the HSQC cross-peaks H6/C6 were attributed to β-d-galactopyranose residues B [17].

^13^C NMR spectrum (Figure 3b) indicated that LJNF3 possessed α-l-fucopyranose residues because of several intense signals in the anomeric (99‒105 ppm) and high-field (16.8‒17.5 ppm) regions. The signal 100.77 ppm was assigned to anomeric C1 carbon, and signals in the region 61‒82 ppm were assigned to the C2–C5 carbons of the pyranoid ring [42]. The carbonyl carbon signal at 174.5 ppm, and corresponding methyl carbon signals at 20.4 and 21.5 ppm, were assigned to O-acetyl groups, which was confirmed with the two-dimensional ^1^H/^13^C NMR correlation spectra HMBC (Figure 3e) [26]. The signals around 62.2 ppm and 67.1 ppm were assigned to non-6-linked and C-6 of 6-linked β-d-galactopyranose residues in LJNF3 [4,5,10,19]. The HSQC and HMBC 2D NMR spectra of LJNF3 were shown in Figure 3 d–e. It was observed that all the correlations of ^1^H/^13^C could be identified as →3-α-l-fucopyranosyl-4-O-sulfate residues [43,44]. According to the NMR spectra, LJNF3 mainly had a structure of 1→3-linked α-l-fucopyranosyl, sulfated group at O-4 positions. Moreover, it also had →6)-β-d-galactopyranosyl-(1→ units. Compared with those of other previous reports [45,46], LJNF3 had a relatively simpler structure, which is of great significance for exploring the relationship between structure and biological activity.

### 2.3. Cell Viability and NO Production to LJNF

On the grounds of the MTT color reduction assay results, LJNF3 stimulated the proliferation of RAW264.7 cells over the entire tested concentration range, from 3.125 to 25 μg/mL, suggesting that LJNF3 had no toxicity to these cells (Figure 4a). Therefore, these dosages were used for the treatment of LJNF3 in the following experiments.

It is well known that NO, as a biological mediator, played an important role in the inflammation response caused by microbes and tumors [47]. In addition, the level of NO increase and overproduction could damage cell macromolecules and further injure the host tissue [48]. As shown in Figure 4b, the NO production of macrophages was notably increased (*p* < 0.05) by LPS induction, while the production of NO treated with LJNF3 decreased (*p* < 0.05) in a dose-dependent manner, and NO was decreased by 40% more than that of the LPS-only treated group, with 25 μg/mL of LJNF3. With the NO inhibition experiment (the data were not shown), the inhibitory effect of LJNF3 on NO was significantly higher than that of LJNFF1 and LJNF2. Therefore, the following experiments focused on the study of the inflammatory mechanism of LJNF3.

According to previous reports, fucoidan in brown seaweed generally plays a good anti-inflammatory effect in 200 or 400 μg/mL [49,50], and the structure of fucoidan is relatively complex [49,50,51]. For example, the fucoidan fraction of *S. japonica* showed the optimal anti-inflammatory activity in concentrations of 50–200 μg/mL [49] and 50–400 μg/mL [50], respectively. Besides, some studies reviewed the biological activities of fucoidan, and considered that the content of fucose, sulfate groups, molecular weight, and position of the glycosidic linkage would affect the biological activities of fucoidan [32,52,53]. Ye et al. (2020) indicated that the purified fucoidan, from 50 to 200 μg/mL (246.4 KDa), with high sulfate group (36.94%) and rich in fucose (24.15%), had better anti-inflammatory activity [49]. Wang et al. (2020) reported that fucoidan, at 50–400 μg/mL (131.5 KDa), with 9.07% sulfate and 47.15% fucose, showed evident anti-inflammatory activity, and it was composed of (1→3)-α-l-fucose residues with abundant branches consisting of α-l-fucose and β-d-galactose residues [50]. Ni et al. (2020) found that fucoidan (F4), at 25 μg/mL (104.3 kDa), with 30.72% sulfate from *S. japonica*, had anti-inflammatory activity and was composed of fucose (79.49%), galactose (16.76%), xylose (1.08%), mannose (1.84%), and rhamnose (0.82%) [51]. From the above researches, fucoidan with a high molecular weight from *S. japonica* had relatively superior biological activity, with a higher content of sulfate groups and fucose. Besides, Murmansk et al. (2020) have studied the anti-inflammatory activity mechanism of fucoidan from *Fucus* vesiculosus, mainly discussing the anti-inflammatory activity of fucoidan by regulating COX-1/2, hyaluronidase, and the MAPK signaling pathway. In addition, it found that a higher content of sulfate group and fucose of fucoidan generally had superior anti-inflammatory activity [54]. Compared with those of the above-reported researches, LJNF3 with a high molecular weight and high content of fucose as a novel galactofucan showed an obvious better anti-inflammatory activity at a relatively lower concentration range of 3.125–25 μg/mL.

### 2.4. Effect of LJNF3 on Pro-Inflammatory Cytokine Secretion

It has been suggested that pro-inflammatory cytokines secreted by macrophages in innate immunity as well as adaptive immune responses, exhibit the abilities to stimulate apoptosis of abnormal cells (infected and transformed), digest harmful microbes, and destroy the extracellular matrix [55]. IL-6 is one of the most important mediators of fever and the acute-phase response [56]. TNF-α has been reported as the key cytokine in immune and inflammatory reactions, and induces the expression of other immunoregulatory mediators [47]. In order to assess the inhibitory effects of LJNF3 on the production of TNF-α (Figure 5a), IL-1β (Figure 5b), and IL-6 (Figure 5c), the levels of protein were determined by ELISA. In Figure 5a, when the cells were exposed to LJNF3 at 25 μg/mL, the release of TNF-α was reduced by 39.17% compared to the group that was only treated with LPS. LJNF3 also exhibited a positive effect on suppressing the expression of IL-1β and IL-6 in the culture medium. The secretion levels of IL-1β and IL-6 from a negative control were measured to be 6.77% and 7.09%, which were significantly increased in LPS-activated macrophage cells. However, the production of IL-6 and IL-1β were dose-dependently decreased by pretreatment with LJNF3. At the highest testing concentration, LJNF3 caused 47.08% and 49.37% decreases in the production of IL-1β and IL-6, respectively. These results were consistent with the effects of some other polysaccharides on the cytokine production by macrophages. Wang et al. (2020) reported that fucoidan from *S. japonica* inhibited the secretion of TNF-α and IL-1β in LPS-activated RAW 264.7 cells, and the inhibition rates were about 48% or 38% at 50 μg/mL, respectively [50]. While LJNF3 could inhibit 39.7% and 47.08% of TNF-α and IL-1β at 25 μg/mL, which exhibited almost the same inhibition activity compared to Wang et al. at 50 μg/mL. It showed that LJNF3 had superior anti-inflammatory activity at a low concentration.

### 2.5. Effect of LJNF3 on iNOS and COX-2 Protein Expression

It is well accepted that iNOS synthesizes a large amount of NO once induced, and causes a significant increase in the concentration of NO [57]. Therefore, the production of NO was mediated by the expression of the iNOS gene. In Figure 6a, the high expression of the iNOS protein in LPS-activated macrophages was determined to be 17.94 ± 1.33 folds. When pretreated with LJNF3 at the concentrations from 3.125 to 25 μg/mL, the expression was observably (*p* < 0.05) decreased in a dose-dependent manner. Researches have shown that excessive COX-2 expression can lead to a variety of chronic diseases, and inhibiting COX-2 overexpression can be an effective way to prevent and treat inflammation and tumors. Furthermore, COX-2 catalyzes the production of prostaglandins (PGs), which represents an important step in the inflammatory process. A dose-dependent inhibition of the protein level of COX-2 was observed (Figure 6b). At 25 μg/mL of LJNF3, the expression fold of COX-2 was determined to be 12.84 ± 1.31, which was decreased to 59.10% compared to that of the LPS-only treated group. For assessing the inhibitory effect of fucoidan on iNOS and COX-2 expression, some research indicated that the protein levels of iNOS and COX-2 were suppressed with pretreatment of 50–200 μg/mL [49]. Compared with the previous research, purity fucoidan did not exert effective anti-inflammatory activity at the concentration of 50 μg/mL, but LJNF3 inhibited the production of iNOS and COX-2 at 25 μg/mL.

### 2.6. Effect of LJNF3 on MAPK and NF-κB Activation

When cells are stimulated with inflammatory factors, mitogen-activated protein kinases (MAPK) are phosphorylated and activate factors that trigger the transcription of inflammatory mediators. In the present researches, MAPK signaling pathways can be activated by LPS, and the inhibition of MAPK phosphorylation is an effective way to cure inflammation [58]. From Figure 7, the phosphorylation of MAPK protein was increased by LPS (*p* < 0.05), obviously. It was found that the content of proteins notably increased, by approximately 3.98-fold (p-ERK), 5.81-fold (p-JNK), and 7.25-fold (p-P38), respectively, compared with the control group (without LPS-treated group). While LJNF3 showed an inhibition of ERK, JNK and P38 phosphorylation, and the inhibitory effect enhanced with the increase in the concentration of LJNF3. When the cells were treated with the highest testing concentration of LJNF3, compared to the LPS-only treated cells, the phosphorylation of ERK (Figure 7a), JNK (Figure 7b), and P38 (Figure 7c) were suppressed to 56.79%, 45.51%, and 52.49%, respectively. In recent years, the mechanisms of the anti-inflammatory effect have been investigated for seaweed extract. The intracellular signaling pathways were suggested to be necessary objects for the activated macrophages. For example, Jaywardena et al. (2020) [59] found that fucoidan can hinder MAPK and NF-κB activation through the dose-dependent down-regulation of pro-inflammatory cytokines.

The transcription factor NF-κB has been considered as the most important transcription factor involved in immunity, inflammatory responses, cell proliferation, and cell death [60]. In a normal condition, NF-κB, a dimer of p65 and p50 subunits, combined with its endogenous inhibitor, IκB, to form an inactive former (p50–p65–IκB). When cells are activated by inflammatory stimulants, such as LPS, the expression of *p*-p65 and *p*-IKKα/IKKβ of the NF-κB signaling pathway were affected. From Figure 7d, it was shown that the phosphorylation level of P65 was increased in LPS-activated macrophages. However, the phosphorylation of P65 in macrophage cells decreased in a dose-dependent manner (*p* < 0.05) when incubated with a tested concentration (3.125‒25 μg/mL) of LJNF3. LJNF3 of 25 μg/mL induced a dramatic decrease in p-P65 expression, from 1.22-fold to 1.04-fold, by LPS activation. IKKα and IKKβ consist of IκB kinases (IKKs), which are responsible for the phosphorylate IκB. During this study, the phosphorylation level of IKKα/IKKβ was determined to be 2.63 ± 0.26-fold in LPS-induced macrophages as a positive control (Figure 7e). The pretreatment of LJNF3 effectively (*p* < 0.05) inhibited, by 37.38% at 12.5 μg/mL, LJNF3 compared to the positive control. All the results demonstrated that LJNF3 had a great potential to inhibit the inflammation reaction via suppressing the phosphorylation of MAPK and NF-κB. Compared to the above reports [49,50], our results indicated that LJNF3 had a positive effect on inhibiting inflammation mediators and cytokines and down-regulating the expression of intracellular signaling pathways at a lower concentration.

### 2.7. Protective Effect of LJNF3 on LPS-Induced Toxicity by Zebrafish Model

In this research, the toxicity test showed that LJNF3 had no toxicity to zebrafish embryos at 2.5–50 μg/mL. Therefore, the study evaluated the protective effect of LJNF3, with 12.5–50 μg/mL, on LPS-induced toxicity in zebrafish embryos. In Figure 8a, the survival rate dramatically decreased to 51.72% by LPS treatment compared with that of the control group, while the rates were significantly increased in a dose-dependent manner after pre-incubating with LJNF3, and the survival rate reached 62.09% at 50 μg/mL of LJNF3. In Figure 8b, LPS treatment significantly increased the heart-beating rate of zebrafish larvae to 117.25% compared with the control group, but the heart-beating rates observably decreased to 105.93% at 50 μg/mL (*p* < 0.05) of LHNF3. Therefore, LJNF3 had an obvious protective effect on zebrafish embryos and could reduce LPS-induced inflammation.

### 2.8. Effect of LJNF3 against LPS-Induced Cell Death, ROS and NO Production

In Figure 9a, strong fluorescence was observed in the positive control group by LPS treatment compared to the control group without LPS, and the cell death rate even reached 175.04%. In the LJNF3 pretreated groups, the fluorescence decreased and showed no obvious difference to the control group (*p* = 0.696) at the concentration of 50 μg/mL. The study indicated that the cell death of zebrafish larvae was inhibited by LJNF3 pretreatment.

The production of ROS is an important factor for confirming the mechanism of inflammatory regulation. As shown in Figure 9b, there was only a small amount of ROS in zebrafish in the control group, while after LPS stimulation, the fluorescence intensity was significantly increased and the ROS content was 2.66-fold that of the control one. However, the ROS production in zebrafish larvae was reduced dose-dependently in pretreatment with LJNF3 compared with the LPS group. Besides, the research evaluated the protective effect of LJNF3 on zebrafish larvae against LPS-activated NO generation. In Figure 9c, the fluorescence intensity of the LPS group was 3.28-fold that of the control group, while the production of NO in zebrafish larvae was reduced in a dose-dependent manner after LJNF3 pretreatment. Compared with the LPS group, the content of NO in zebrafish larvae was the lowest at 50 μg/mL, and the inhibition rate was about 59.43% (*p* < 0.05). Sanjeewa et al. (2019) reported that the purified fucoidan at 12.5–50 μg/mL showed a protective effect against cell damage in LPS-induced zebrafish larvae, while suppressing the NO production caused by LPS [61]. According to the above reports, it revealed that LJNF3 had a potential anti-inflammatory effect because of its low cytotoxicity and inhibition effect at a lower concentration.

## 3. Materials and Methods

### 3.1. Materials and Reagents 

*S. japonica* was harvested on 20 June 2019 in Rongcheng, Shandong province, China. The strain is named Yanza, which is the traditionally hatched and cultured strain in Rongcheng *S. japonica* cultural area. Diaminofluorophore 4-amino-5-methylamino-2′7′-difluorofluorescein diacetate (DAF-FM DA), 2, 7-dichlorodihydrofluorescein diacetate (DCFH-DA), and acridine orange were from Sigma company (St. Louis, MO, USA). The monosaccharide standards of D-fucose (Fuc), galactose (Gal), L-arabinose (Ara), D-mannose (Man), L-d-rhamnose (Rha), D-glucose (Glc), and D-xylose (Xyl), as well as Griess reagents, lipopolysaccharide (LPS), dimethyl sulfoxide (DMSO), and 3-(4,5)-dimethylthiazol-2-yl)-2,5-diphenyltetrazolium bromide (MTT) were also obtained from Sigma Chemical Co. (St. Louis, MO, USA). Fetal bovine serum (FBS) and Dulbecco’s modified Eagle’s medium (DMEM) were from company of Gibco (Rockville, MD, USA). ELISA kits were purchased from Dakewei (Shenzhen, China).

### 3.2. Extraction and Purification of Polysaccharides

*S. japonica* was washed with tap water followed by distilled water to remove sand and salt on the surface. The cleaned sample was air-dried in an oven at 60 °C. The dried sample was grounded to powder, and the fractions smaller than a size of 60 mesh were collected and stored in −20 °C until use. In order to remove the polyphenols, pigments and lipids, the sample was extracted with 95% ethanol at 40 °C for 2 h. The residue was collected by filtration. The defatted sample was incubated with ultrapure water (1:30, *w*/*v*) at 120 °C for 2 h, and the supernatant was obtained by centrifugation (2000× *g* for 20 min). Then, 2% CaCl_2_ solution was added to the supernatant to remove alginate and the derived alginate precipitate was removed by centrifugation at 2000× *g* for 15 min. The supernatant was concentrated by a rotary evaporator, dialyzed against distilled water and lyophilized by a freeze-drier to obtain the crude LJN.

By anion-exchange chromatography, the crude LJN was purified by AKTA purifier [51]. Three fractions (LJNF1, LJNF2, LJNF3) were pooled based on the total carbohydrate content by phenol–sulfuric acid method [62]. According to the references, determination of sulfated content, carbohydrate content, phenol content and protein content of each fraction were studied [62,63,64,65].

### 3.3. Infrared Spectroscopic Analysis

Nicolet 67 spectrometer (Thermo Scientific, Waltham, MA, USA) was used to analyze LJNF3 in the range of 400–4000 cm^−1^.

### 3.4. Molecular Weight Analysis

The chromatographic column was TSK Gel GMPWXL (300 mm × 7.8 mm). DAWN HELEOS-II 18 angle laser detector and differential detector made by Wyatt Company (Santa Barbara, CA, USA).

### 3.5. Monosaccharide Composition Analysis

The monosaccharide composition was analyzed according to the reference reported by previous report using 6890 N gas chromatographic system (Agilent 6890 N, Agilent, Palo Alto, CA, USA) [56].

### 3.6. NMR Spectroscopy

Agilent DD2 500 MHz (Agilent, Palo Alto, CA, USA) recorded on NMR spectra (^1^H NMR, ^13^C NMR, COSY, HSQC and HMBC).

### 3.7. Cell Culture and Cell Viability Analysis

RAW 264.7 macrophage cells (MFN048; FDCC, Shanghai, China) were cultured at 37 °C in a humidified incubator (HERAcell 150 i, Thermo, Waltham, MA, USA) with 5% CO_2_ in DMEM supplemented with 1% penicillin–streptomycin and 10% fetal bovine serum [50,51]. The effects of LJNF3 with 3.125, 6.25, 12.5 and 25 μg/mL on the cell viability were determined by MTT assay at 490 nm [66].

### 3.8. Inflammatory Responses Analysis

RAW 264.7 cells were pretreated with various concentrations of LJNF3 for 1 h and then stimulated with or without LPS (1 μg/mL) for 24 h. The supernatants were collected for the analysis of pro-inflammatory cytokines release. The TNF-α, IL-1β, and IL-6 were determined by ELISA kits (Dakewei, Shenzhen, China) [50,51].

### 3.9. Western Blot Analysis

The Western blot was analyzed according to the reference [51].

### 3.10. Origin and Maintenance of Parental Zebrafish

The sources and cultivation methods of adult zebrafish are referred to by Ni et al. [51]. The embryos used in the experiment were obtained from natural mating of healthy male and female zebrafish. This experiment was approved by the ethical committee of experimental animal care at the College of Food Science and Engineering at the Ocean University of China (approval No. 2019-05).

### 3.11. Survival Rate and Heart-Beating Rate Analysis

Survival rate and heart-beating rate were measured by the method described by reference [67].

### 3.12. Inflammation-Induced Intracellular Cell Death, ROS and NO and Image Analysis

Intracellular cell death, ROS, and NO in zebrafish embryos were estimated according to the method by Zou et al. [68].

### 3.13. Statistical Analysis

All experiments were carried out in independent triplicates, and each experiment had at least three parallel groups. All results were expressed as the mean ± standard error. Significant differences were determined via IBM’s SPSS (version 19.0, SPSS Inc., Chicago, IL, USA) statistics by one-way analysis of variance (ANOVA). Values with *p* < 0.05 were significantly different. Different letters indicate a significant difference between the compared pairs of individual means, while the same letter means no significant difference between the compared pairs of individual means [50,51].

## 4. Conclusions

It is the first time discovering LJNF3 as galactofucan extracted from *S. japonica*, and evaluating the anti-inflammatory effect in vitro and vivo. The FTIR and NMR analysis demonstrated that LJNF3 contained a main chain of (1→3)-α-l-fucopyranosyl residues, with sulfate groups mainly at the C4 positions, and 6-linked β-d-galactopyranose units. The anti-inflammatory effect of LJNF3 was measured in vitro by RAW 264.7 cells. LJNF3 had no toxicity at the concentration of 3.125‒25 μg/mL in macrophages, and further suppressed the production of NO, TNF-α, IL-1β, and IL-6 in a dose-dependent manner, which was related to the translational down-regulation of iNOS, COX-2, MAPK, and NF-κB signaling pathways. Moreover, it possessed the protective effect of LJNF3 by zebrafish, through decreasing the cell death and production of ROS and NO. This study provides scientific evidence and advances in the potential application of LJNF3 from *S. japonicas*, an attractive anti-inflammatory agent for biomedical, food and cosmetic industries.

## Figures and Tables

**Figure 1 marinedrugs-19-00430-f001:**
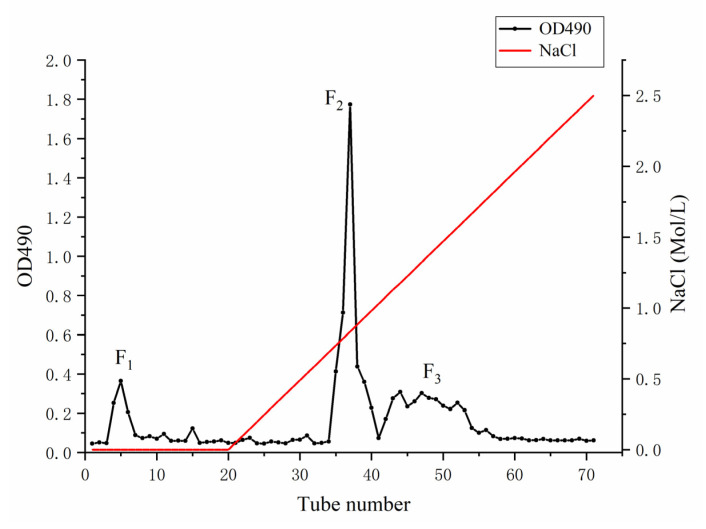
DEAE-Sepharose Fast Flow anion-exchange chromatography for the purification of LJN.

**Figure 2 marinedrugs-19-00430-f002:**
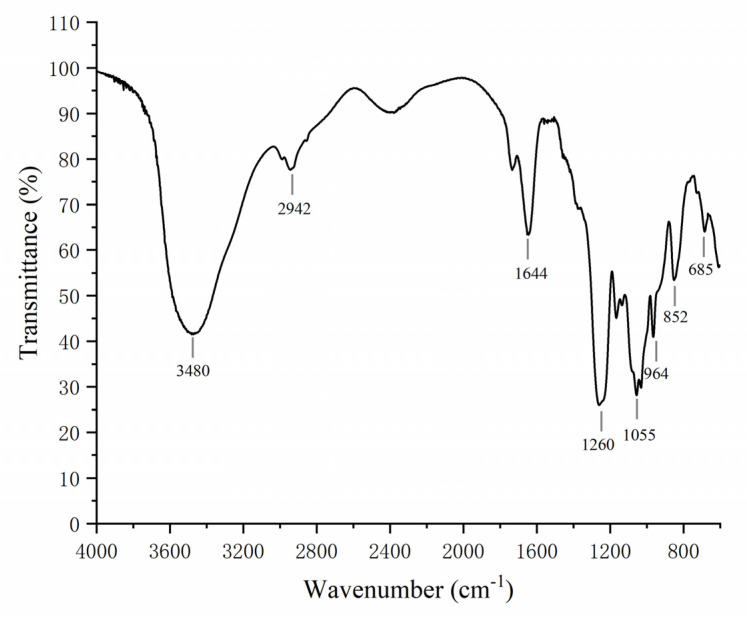
FTIR spectrum analysis of LJNF3.

**Figure 3 marinedrugs-19-00430-f003:**
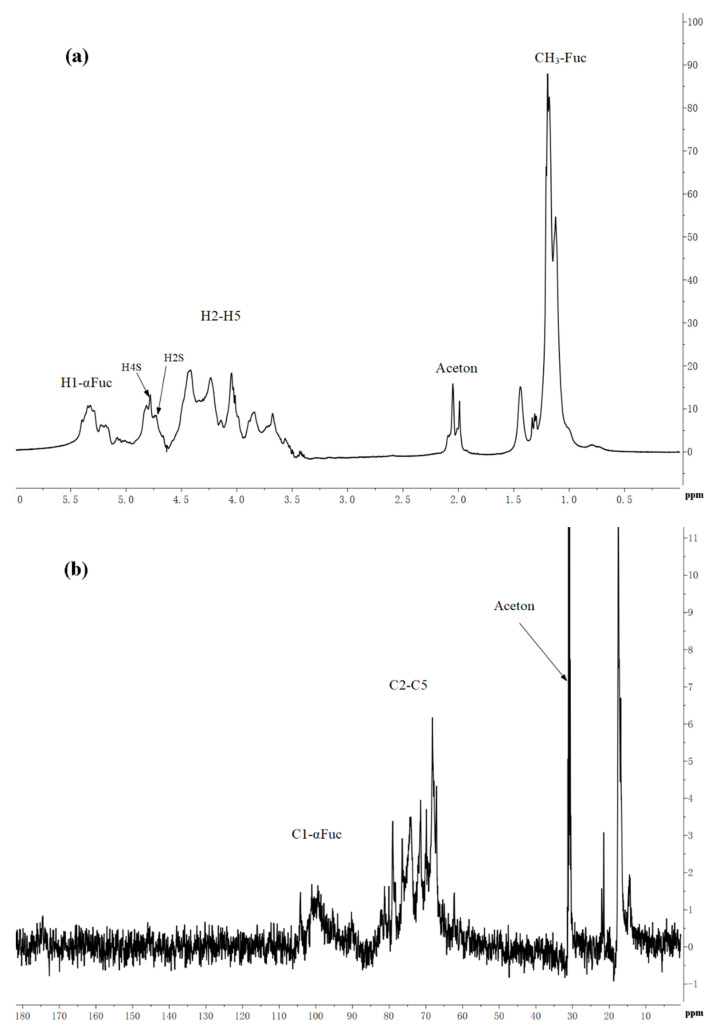
The ^1^H (**a**), ^13^C (**b**), COSY (**c**), HSQC (**d**) and HMBC (**e**) spectrum of LJNF3.

**Figure 4 marinedrugs-19-00430-f004:**
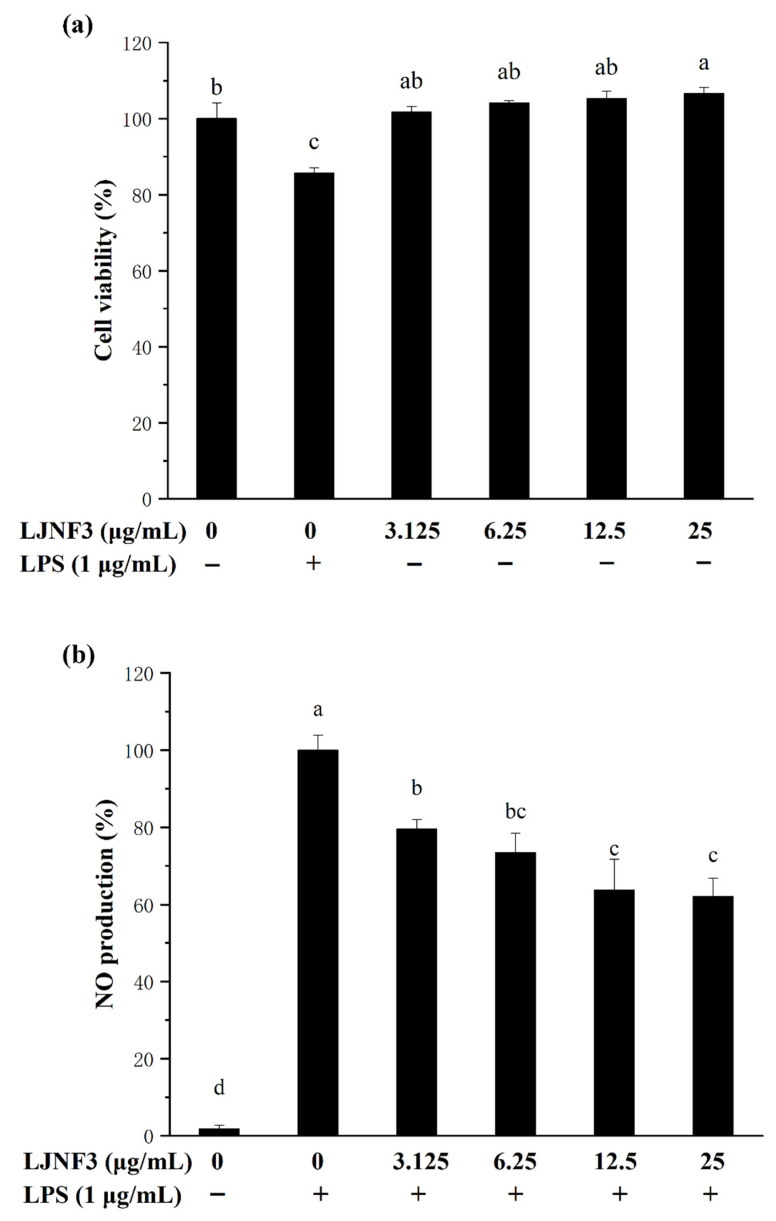
Effect of LJNF3 on RAW 264.7 cell viability (**a**) and NO production (**b**). Values correspond to mean ± SD of three independent experiments. Different letters indicate significant difference (*p* < 0.05) via Tukey’s multiple range test.

**Figure 5 marinedrugs-19-00430-f005:**
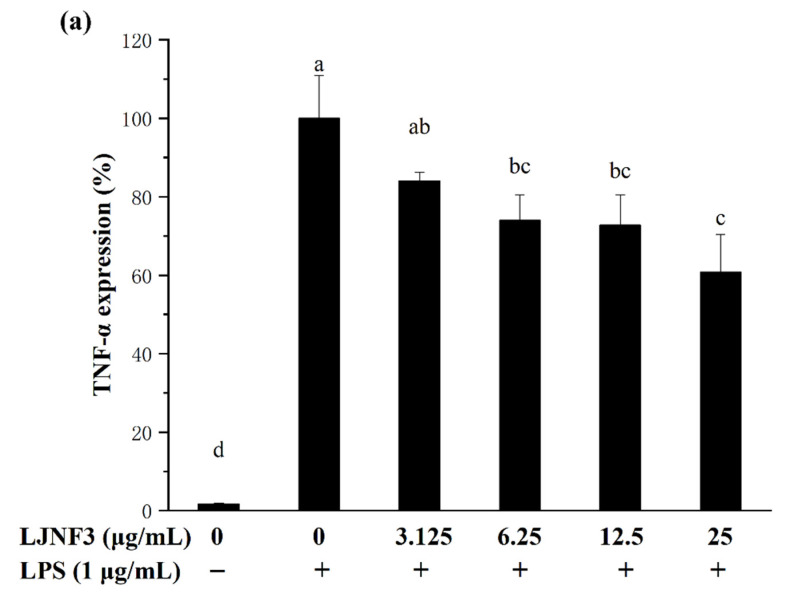
Effect of LJNF3 on TNF-α (**a**), IL-1β (**b**) and IL-6 (**c**) protein levels in LPS-stimulated RAW 264.7 cells. Values correspond to mean ± SD of three independent experiments. Different letters indicate significant difference (*p* < 0.05) via Tukey’s multiple range test.

**Figure 6 marinedrugs-19-00430-f006:**
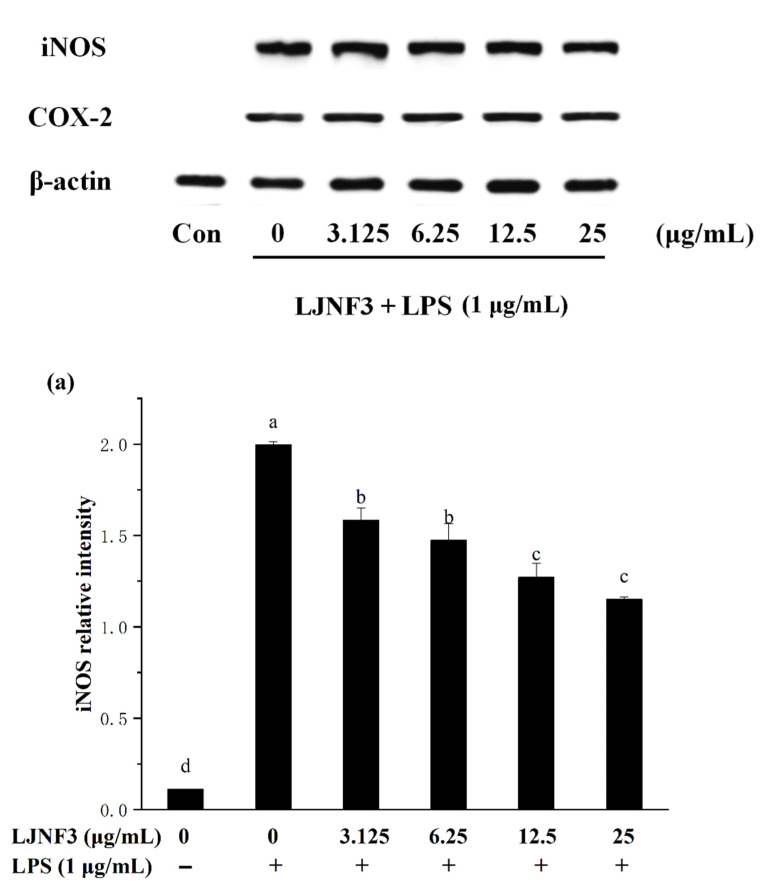
Effects of LJNF3 on the expression of iNOS (**a**) and COX-2 (**b**) in LPS-induced RAW 264.7 cells. Values correspond to mean ± SD of three independent experiments. Different letters indicate significant difference (*p* < 0.05) via Tukey’s multiple range test.

**Figure 7 marinedrugs-19-00430-f007:**
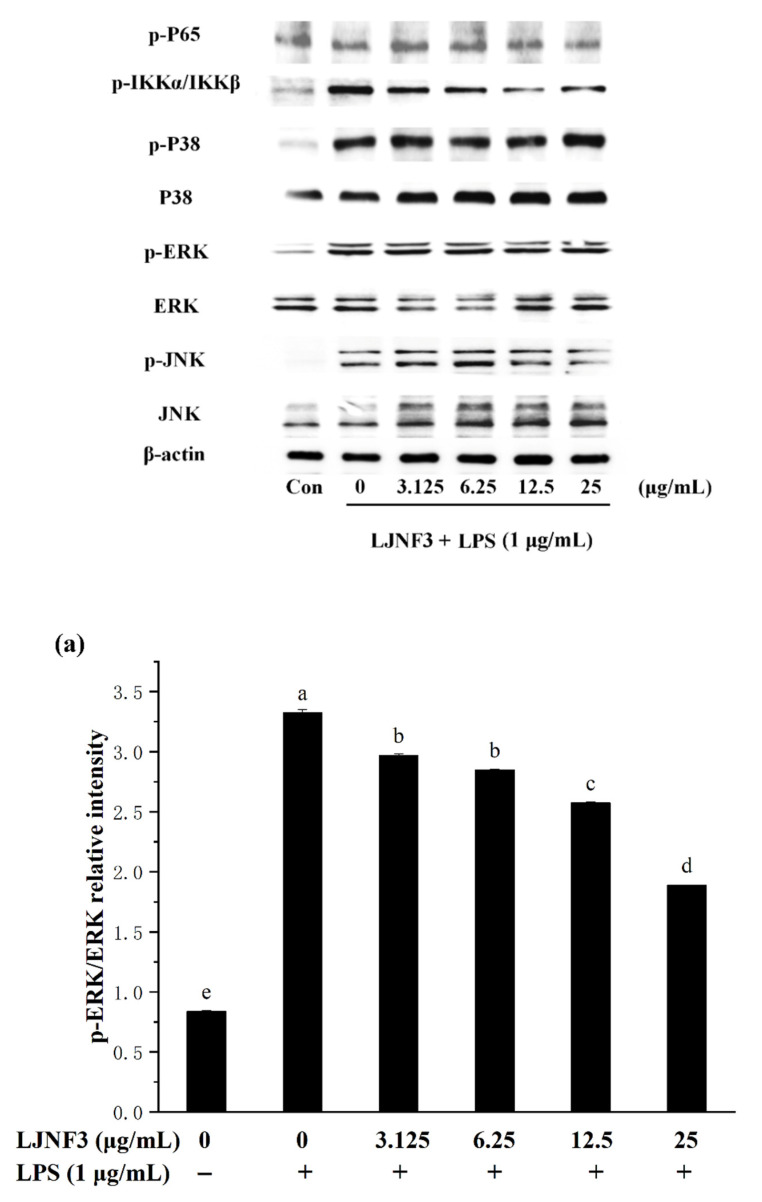
Effects of LJNF3 on the MAPK and NF-κB signaling pathways in LPS-stimulated RAW 264.7 cells. (**a**) p-ERK/ERK; (**b**) p-JNK/JNK; (**c**) p-p38/P38; (**d**) p-P65; and (**e**) p-IKKα/IKKβ. Values correspond to mean ± SD of three independent experiments. Different letters indicate significant difference (*p* < 0.05) via Tukey’s multiple range test.

**Figure 8 marinedrugs-19-00430-f008:**
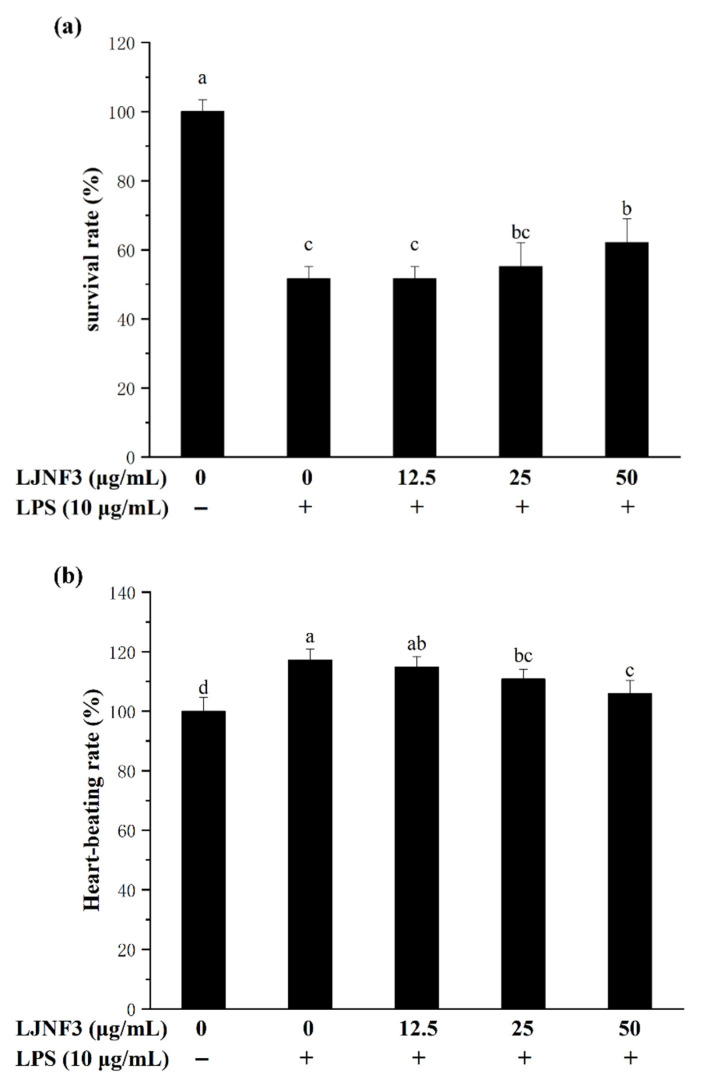
Effects of LJNF3 on LPS-induced survival rate (**a**) and heart-beating rate (**b**) in zebrafish embryos. Values correspond to mean ± SD of three independent experiments. Different letters indicate significant difference (*p* < 0.05) via Tukey’s multiple range test.

**Figure 9 marinedrugs-19-00430-f009:**
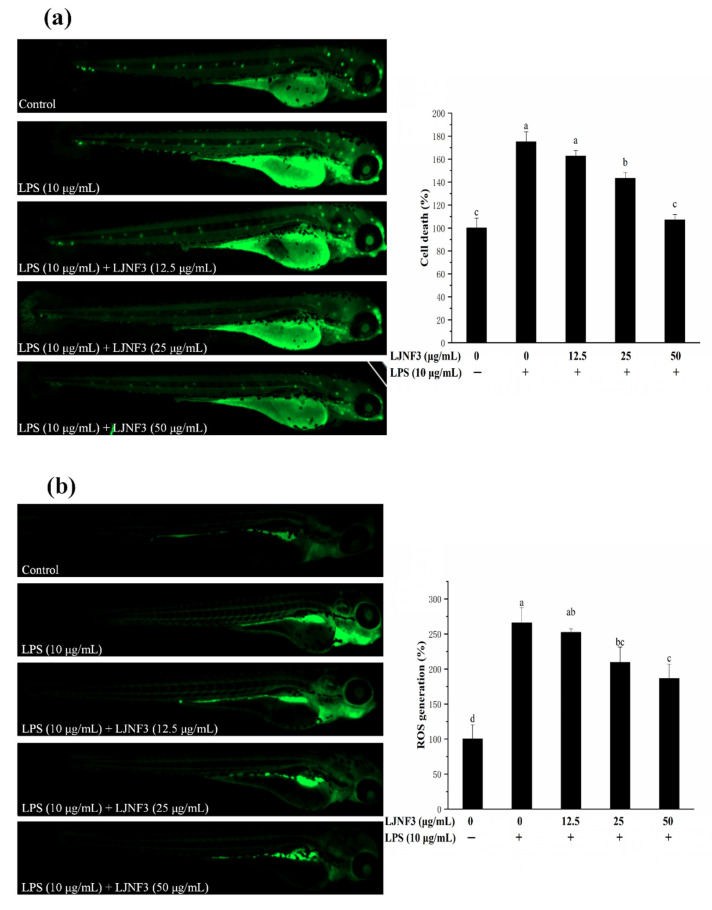
Inhibitory effects of LJNF3 on LPS-induced cell death (**a**), ROS production (**b**), and NO production (**c**) in zebrafish embryos. Values correspond to mean ± SD of three independent experiments. Different letters indicate significant difference (*p* < 0.05) via Tukey’s multiple range test.

**Table 1 marinedrugs-19-00430-t001:** Chemical composition, monosaccharide composition and molecular weight of LJNF1, LJNF2 and LJNF3.

Sample	Total Sugar (%)	Sulfate (%)	Phenol (%)	Protein (%)	Monosaccharide Composition (%)						Molecular Weight (kDa)
					Rhamnose	Fucose	Xylose	Mannose	Galactose	Glucose	
LJNF1	71.7 ± 0.25	9.31 ± 0.33	0.05 ± 0.01	0.07 ± 0.02	0.95	39.97	3.76	10.41	39.24	5.67	2113
LJNF2	75.58 ± 0.53	8.42 ± 0.44	0.12 ± 0.01	ND	1.56	45.67	7.41	21.79	15.03	8.54	324.3
LJNF3	69.12 ± 0.94	11.36 ± 0.42	ND	ND	ND	79.22	ND	ND	20.78	ND	261.7

ND: not detected.

**Table 2 marinedrugs-19-00430-t002:** Monosaccharide composition, sulfate group, molecular weight and biological activity of galactofucan in brown algae in recent years.

	Source of Algae	Monosaccharide Composition	Sulfate Group	Molecular Weight	Biological Activities	Reference
*Saccharina (Laminaria)*	*Saccharina (Laminaria) japonica*	Fuc:Gal = 79.2:29.78	11.36 ± 0.42%	261.7 KDa	Anti-inflammatory activity	this study
	*Saccharina (Laminaria) japonica*	Fuc:Gal = 1:0.25	21%	13.7 KDa	Antiviral activity	[2]
	*Saccharina (Laminaria) japonica*	Fuc:Gal = 1:0.26	48.26%	ND	Growth-promoting activity of fibrocytes	[3]
	*Saccharina gurjanovae*	Fuc:Gal = 76.3:23.7 mol%	25.1%	71 KDa	Anticancer	[7]
	*Saccharina* *latissima*	Fuc:Gal:Xyl = 86.1:11.1:2.8	ND	416,000	Antitumour	[7]
	*Saccharina japonica*	Fuc:Gal:Man:Xyl = 49.9:44.1:5.3:1.1	23.2%	1800 KDa	Antiviral activity	[33]
	*Laminaria japonica*	FucXyl:Gal:Glc:GalUA:Man = 14.9:1.0:16.8:1.7:3.9:6.3	26.7%	527.3 KDa	Modify the intestinal microbiota	[1]
	*Saccharina sculpera*	Fuc:Gal:Glu:Man:Rha:Xyl:GlcA = 16.67:31.90:2.50:6.36:1.46:2.20:6.82	27.13 ± 0.79%	527.3 KDa	Hypolipidemic effect	[8]
	*Laminaria japonica*	Fuc:Gal:Man:Glc:Rha = 1:0.172:0.016:0.015:0.003	41.80%	ND	Neuron protective effect	[6]
	*Laminaria japonica*	Fuc:Gal:Man:Glc:Rha = 81.09:15.31:1.42:1.35:0.23	41.80%	8.1 KDa	ND	[5]
*Sargassum*	*Sargassum feldmannii*	Fuc:Gal = 72:28 mol%	25.3%	237.7 KDa	Anticancer	[9]
	*Sargassum duplicatum*	Fuc:Gal = 51:49 mol%	31.7%	191 KDa	Anticancer	[9]
	*Sargassum mcclurei*	Fuc:Gal = 58.5:41.5 mol%	35%	ND	Anticancer	[10]
	*Sargassum henslowianum*	Fuc:Gal = 3:1	31.9%	6.55 × 10^5^	Antiviral activity	[11]
	*Sargassum fusiforme*	Fuc:Gal = 59.6:44.4 mol%	ND	ND	Anti-angiogenic activity	[12]
	*Sargassum thunbergii*	Fuc:Gal = 1:0.46	38.79%	121.2 KDa	Anticancer	[13]
	*Sargassum polycystum*	Fuc:Gal:Xyl = 36.0:19.1:1.7	33.7%	ND	ND	[18]
	*Sargassum thunbergii*	Man:Rha:GlcA:Glc:Gal:Xyl:Fuc = 0.04:0.03:0.05:0.04: 0.41:0.02:1.00	23.01%	143.0 and 36.7 KDa	Anti-tumor and Anti-angiogenic activities	[17]
	*Sargassum thunbergii*	Man:Rha:GlcA:Glc:Gal:Xyl:Fuc = 0.59:0.08:0.31:0.04:0.47:0.08:1.00	14.81%	135 KDa	Neuroprotective activities	[14]
	*Sargassum fusiforme*	Fuc:Xyl:Gal:GlcA:Man = 1:0.03:0.24:0.02:0.02	45.02%	151.2 KDa	ND	[16]
*Alaria*	*Alaria angusta*	Fuc:Gal = 52.6:47.4 mol%	24.0%	ND	Anticancer	[19]
	*Alaria marginata and A. angusta*	Fuc:Gal:Xyl = 47.5:47.3:5.2	28.3%	ND	Anticancer	[20]
*Azolla*	*Azolla filiculoides?*	Fuc:Gal = 61.25:38.75	ND	992.9 × 10^3^ g/mol	Immunological activity	[34]
*Hizikia*	*Hizikia* *fusiforme*	Fuc:Gal:Xyl = 1:0.27:0.01	39.85%	99.2 KDa	Anti-complement activity	[22]
*Hormophysa*	*Hormophysa cuneiformis*	Fuc:Gal:Xyl = 39.3:9.6:1.0	35.3%	121 KDa	Anticoagulant	[23]
*Lobophora*	*Lobophora* *variegata*	Fuc:Gal:Xyl = 29.2:36.8:0.1	ND	35 KDa	Anti-oxidant and anti-inflammatory	[25]
*Padina*	*Padina boryana*	Fuc:Gal:Man:Glu = 39.8:36.7:17.4:6	18.6%	317.5 and 8.5 KDa	Anticancer	[35]
*Undaria*	*Undaria pinnatifida*	Fuc:Gal:Xyl:Man = 50.9:44.6:4.2:0.3	ND	1246 KDa	Antitumor	[26]
	*Undaria pinnatifida*	Fuc:Gal:Rha = 54:45:1	ND	290 KDa	Antiviral activity	[27]
*Scytosiphon*	*Scytosiphon lomentaria*	Fuc:Gal = 88:12 mol%	29.5%	8.5 KDa	Antiviral activity	[28]
*Spatoglossum*	*Spatoglossum* *schroederi*	Fuc:Gal:Xyl = 1.0:2.0:0.5	36.36%	21.5 KDa	Hemostatic Activities	[29]
	*Spatoglossum schröederi*	Fuc:Gal:Xyl = 1.0:2.0:0.5	15.0%	21.5 KDa	Antitumor	[30]
*Costaria*	*Costari* *a costata*	Fuc:Gal:Man:GlcA = 70.2:19.8:7:3	23.8%	160 KDa	ND	[21]
*Eclonia*	*Eclonia cava?*	Fuc:Gal:Man:Rha:Xyl:Glc = 1:0.83:0.01:0.05:0.06	18.9%	ND	Anticancer	[36]
*Kjellmaniella*	*Kjellmaniella crassifolia*	Fuc:Gal:Man:Xyl:Glc:GlcA = 1:0.35:0.05: 0.03:0.01:0.06	32.5%	258 KDa	Antitumor	[24]

ND: not detected.

**Table 3 marinedrugs-19-00430-t003:** ^1^H and ^13^C NMR data for LJNF3.

Residue	Name	^1^H and ^13^C Chemical Shifts (ppm)					
A	→3)-α-l-Fucp-4-SO_3_−(1→	H-1	H-2	H-3	H-4	H-5	H-6
		5.05	3.93	4.52	4.25	3.97	1.20
		C-1	C-2	C-3	C-4	C-5	C-6
		100.77	65.56	72.36	80.08	66.85	19.97
B	→6)-β-d-Galp-(1→	H-1	H-2	H-3	H-4	H-5	H-6
		4.46	3.55	3.72	3. 98	3.89	3.84
		C-1	C-2	C-3	C-4	C-5	C-6
		104.25	72.77	73.70	69.85	74.21	65.75

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
