# Peer review of "Molecular Mechanism of Anti-Inflammatory Activities of a Novel Sulfated Galactofucan from Saccharina japonica"

_marinedrugs, 2021, doi:10.3390/md19080430_

Round 1

Reviewer 1 Report

Increasing knowledge of the mechanisms of brown algae compounds is extremely important. However, while reading the manuscript, I had some questions and comments.

  1. In section 3.1, please indicate the time of collection for S. japonica. Indicate who identified the alga.
  2. How was the yield of purified polysaccharides F1-F3 calculated?
  3. Please transfer Table 2 to the Discussion section.
  4. Galactofucan of the composition isolated from fucose fucose, glucose, galactose, xylose, mannose, and arabinose 1.0: 0.16: 0.05: 0.09: 0.03: 0.03 (https://doi.org/10.3390/md16040132) showed a pronounced anti-inflammatory effect (https: / /doi.org/10.3390/md18050275). Please add information on the molecular weight of the isolated galactofucans to Table 2. Discuss the influence of a complex of factors, including molecular weight, on the activity of these compounds.
  5. Reference [33] does not provide information on the absorption of the polysaccharide with a molecular weight of 261.7 kDa. Please provide the correct reference.
  6. Please explain on the basis of what data only one polysaccharide was selected for further study. Why didn't you study the F2 polysaccharide with a similar molecular weight?
  7. If using the standard procedures provided by the kit manufacturers, please provide a reference to those procedures. If you used your methods, please provide validation parameters for all tests. Indicate the statistical difference in Figures 4-9.
  8. In the Abstract section, please include experimental data confirming your sentences.
  9. Please discuss in more detail why the positive effect of F3 fraction is related to previously published data (Ye, J .; Chen, D .; Ye, Z. et al. Fucoidan Isolated from Saccharina japonica Inhibits 548 LPS-Induced Inflammation in Macrophages via Blocking NF-kappaB, MAPK and JAK-STAT Pathways. Marine Drugs 2020, 18, 549).

Reviewer 2 Report

The paper analyzes the chemical structure and the bioactivity of one fraction of Saccharina japonica extract. It lacks a bit in novelty and originality, being the work done and the structure almost the same as the article already published by the same authors (Ni et al. 2020 In vitro and in vivo anti-inflammatory activities of a fucose-rich fucoidan isolated from Saccharina japonica https://doi.org/10.1016/j.ijbiomac.2020.04.012). Still, a serious research effort has been made, therefore I do think it deserves to be published, but not in the present form indeed.

The main problem is represented by the English. The manuscript must be profoundly revised. There are plenty of translation errors. I suggest asking a native speaker or consult a translation agency. Aside from the grammar problem, the text must be rewritten to make it more fluid.

In particular, the introduction must be rewritten and expanded. More literature on the anti-inflammatory effect of these compounds should be included and discussed.

Minor comments:

Please, explain why all the analyses were carried out only on the third fraction of the polysaccharide extract

Most of the figures are of low quality, at the limit of readability. Please fix them.

Please explain the statistical results of the Tukey’s (and not Turkey’s, correct it) post-test. Specify the meaning of each letter, it is not clear.

As regards the ELISA, please report the results as concentration, and not as a percentage of variation

Also, comparing the results found in the previous paper (Ni 2020), why the fractions obtained from the same species of seaweed are different? Please, explain

Reviewer 3 Report

The manuscript is on the anti-inflammatory activities of sulfated galactofucan. The experiments were well conducted.

Before the publication, some modification will be necessary.

  1. There is no exact reason for selecting LJNF3 for further study.  As authors mentioned, the biological activites were dependent on the structures of sulfated galactofucan. The biological activities of other fractions (LJNF1 and LJNF2) are necessary.
  2. It is better to add Table 1 as  supplemetary Table.
  3. In discussion, comparison with other fucoidan samples  should be added with the treated amount.

Round 2

Reviewer 1 Report

The authors wrote that they made the necessary corrections on Q4 (Galactofucan of the composition isolated from fucose fucose, glucose, galactose, xylose, mannose, and arabinose 1.0: 0.16: 0.05: 0.09: 0.03: 0.03 (https://doi.org/10.3390/md16040132) showed a pronounced anti-inflammatory effect (https: / /doi.org/10.3390/md18050275). Please add information on the molecular weight of the isolated galactofucans to Table 2. Discuss the influence of a complex of factors, including molecular weight, on the activity of these compounds).

Indicate the statistical difference in Figures 4-9.

However, I did not find them in the text. Please attach the corrected file. 

Round 3

Reviewer 1 Report

Dear authors, I want to draw your attention to the fact that the anti-inflammatory activity and the mechanism of action of galactofucan isolated from Fucus vesiculosus have been described (https: / /doi.org/10.3390/md18050275). Please compare the obtained data and discuss what the activity of higher galactofucans is related to in comparison with that described in the literature. I did not find any discussion about the relationship of structure and biological activities (line 101-109). You included one phrase about the ineffectiveness of very low molecular weight fucoidans (ref.32). 

Round 4

Reviewer 1 Report

The authors have made the necessary corrections and I have no more questions.